# SegRGB-X: General RGB-X Semantic Segmentation Model

## Abstract

Semantic segmentation across arbitrary sensor modalities faces significant challenges due to diverse sensor characteristics, and the traditional configurations for this task result in redundant development efforts. We address these challenges by introducing a universal arbitrary-modal semantic segmentation framework that unifies segmentation across multiple modalities. Our approach features three key innovations: (1) Modality-aware CLIP (MA-CLIP), which provides modality-specific scene understanding guidance through LoRA fine-tuning; (2) complementary learnable prompts for capturing fine-grained features; and (3) a Modality-aware Selective Adapter (MASA) for dynamic feature adjustment. Evaluated on five diverse datasets with different complementary modalities (event, thermal, depth, polarization, and light field), our model surpasses specialized multi-modal methods and achieves state-of-the-art performance with a mean IoU of 65.03.

## 1 Introduction

The rapid advancement of sensor technologies has significantly promoted progress in multi-modal fusion for semantic segmentation Zhang et al. (2023a;b), generating increasing interest in leveraging diverse sensor modalities to improve segmentation accuracy.

Recent methods Zhang et al. (2023a;b); Jia et al. (2024a); Li et al. (2024) have achieved impressive results across various multi-modal semantic segmentation tasks. However, these approaches primarily rely on modality-specific specialist models, each customized for a particular modality combination. Consequently, a separate model must be trained for every modality pair, leading to redundancy in both model design and training. Moreover, since the data for each task is often limited, such specialist models risk overfitting to dataset-specific distributions and sacrificing generalization ability. While expanding the dataset could mitigate this issue, collecting and annotating multi-modal data is both labor-intensive and time-consuming. To address these limitations, developing a generalist model capable of jointly handling diverse modalities within a single unified architecture presents a promising direction. Such a model can exploit shared representations across modalities and leverage the full available data.

Moreover, large-scale pretrained vision-language models (VLMs), such as CLIP Radford et al. (2021), ALIGN Jia et al. (2021), and BLIP Li et al. (2022),

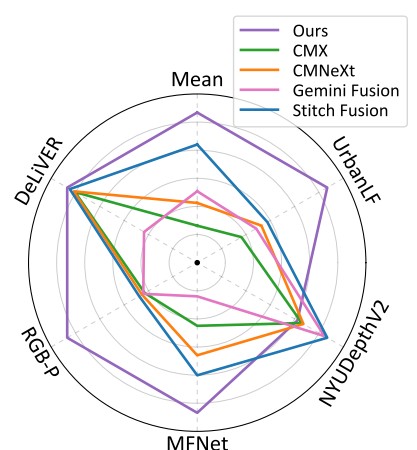

Figure 1: Performance comparison between our method and CMX Zhang et al. (2023a), CMNeXt Zhang et al. (2023b), Gemini Fusion Jia et al. (2024a), and Stitch Fusion Li et al. (2024) on five multi-modal semantic segmentation datasets: DeLiVER Zhang et al. (2023b), MFNet Ha et al. (2017), NYUDepthV2 Silberman et al. (2012), ZJU RGB-P Xiang et al. (2021), and UrbanLF Sheng et al. (2022). Our general model, SegRGB-X, achieves the best overall performance.

have demonstrated strong generalization capabilities across a wide range of vision tasks. These mod-

els leverage massive image-text pairs to learn rich feature representations, making them effective for various downstream applications. However, their impact on multi-modal semantic segmentation remains limited, primarily because they are pretrained on standard RGB images paired with textual descriptions. As a result, their ability to generalize to diverse sensor modalities—such as event, thermal, depth, polarization, or light field—is constrained, reducing their effectiveness in more complex multi-modal settings.

In this work, we propose SegRGB-X, a general RGB-X semantic segmentation model designed to handle diverse sensor modalities. The architecture comprises a backbone equipped with modality-aligned embeddings and a Domain-Specific Refinement Module (DSRM), a Modality-Aware CLIP (MA-CLIP), and a segmentation head. To overcome the limitations of using VLMs for multi-modal segmentation, we fine-tune CLIP on multi-modal segmentation data using LoRA Hu et al. (2022), allowing MA-CLIP to serve as a modality information provider. To mitigate the feature gap between the input embeddings and the control prompts generated by MA-CLIP, we introduce a modality-aligned embedding mechanism with learnable prompts. In the final stage of the backbone, DSRM is developed to refine modality-specific features. As shown in Fig. 1, we evaluate SegRGB-X through joint training on five multi-modal segmentation datasets. Compared with state-of-the-art (SOTA) methods, our model achieves the highest average performance across DeLiVER Zhang et al. (2023b), MFNet Ha et al. (2017), NYUDepthV2 Silberman et al. (2012), RGB-P Xiang et al. (2021), and UrbanLF Sheng et al. (2022).

Our main contributions are summarized as follows:

- We propose SegRGB-X, a general model capable of handling multiple sensor modalities (event, thermal, depth, polarization, and light field) within a single framework, addressing the limitations of modality-specific specialist models.

- We introduce MA-CLIP, which fine-tunes CLIP with LoRA on multi-modal segmentation data, effectively bridging the gap between vision-language pretraining and multi-modal segmentation.

- We design a modality-aligned embedding mechanism that incorporates learnable prompts to align the feature space between input embeddings and control prompts generated by MA-CLIP. In the final stage of the backbone, we develop the DSRM to adaptively refine modality-specific features, thereby enhancing the segmentation performance.

- We conduct joint training and evaluation on five diverse multi-modal semantic segmentation datasets. SegRGB-X achieves a SOTA performance with an average mIoU of 65.03%, outperforming previous specialist models.

## 2 RELATED WORK

**Multi-modal semantic segmentation.** To overcome the limitations of RGB images, recent research in multi-modal semantic segmentation has explored diverse modality combinations to enhance performance beyond traditional RGB-based approaches. RGB-D fusion Qian et al. (2021); Zhou et al. (2022a); Cao et al. (2021) leverages depth information, while RGB-thermal methods integrate thermal-specific fusion strategies Sun et al. (2019; 2020); Zhou et al. (2021). New modalities have emerged, such as polarization cues for transparent object segmentation Kalra et al. (2020) and event-based data for accident scene analysis Zhang et al. (2021a); Cao et al. (2024). Additional advancements include perception-aware fusion for LiDAR data Zhuang et al. (2021), depth-adaptive convolution techniques Wang & Neumann (2018); Xing et al. (2020); Wu et al. (2020), and multi-task masked autoencoding Bachmann et al. (2022). Attention-based fusion methods have also been developed to facilitate cross-modal interaction Zhang et al. (2019); Hu et al. (2019); Zhang et al. (2021b). CMX Zhang et al. (2023a) was a significant work for arbitrary fixed RGB-modality pairs. CMNeXt Zhang et al. (2023b) extended this capability to arbitrary fixed modality tuples. Building upon CMNeXt, several enhanced approaches have emerged. Gemini Fusion Jia et al. (2024a) employs per-pixel attention inspired by token fusion strategies. MAGIC Zheng et al. (2024c) introduces a multi-modal aggregation module to extract complementary scene information efficiently, and its improved variant, Magic++ Zheng et al. (2024a), integrates a Multi-scale Arbitrary-modal Selection Module (MASM) and consistency training. StitchFusion Li et al. (2024) adopts a multi-directional MLP to enhance information sharing and fusion across modalities. Any2Seg Zheng et al. (2024b)

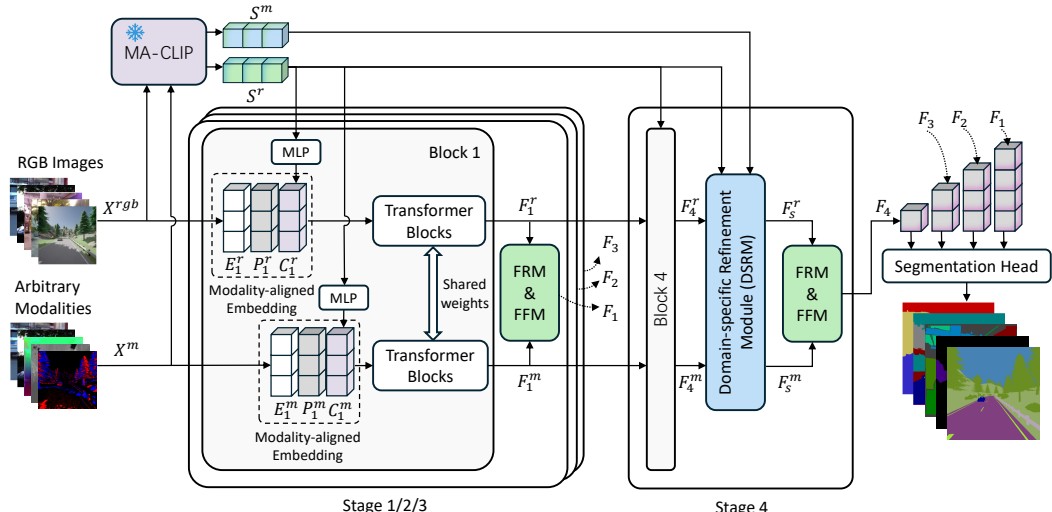

Figure 2: Overall framework of our SegRGB-X model. At each stage, feature embeddings from the MA-CLIP are incorporated into the modality-aligned embedding to enhance feature representations. The input embeddings are processed using shared-weight Transformer blocks Xie et al. (2021), enabling consistent and efficient feature extraction across modalities. The extracted features are progressively fused through the FRM and FFM modules, as introduced in Zhang et al. (2023b). In the final stage, a Domain-Specific Refinement Module (DSRM) is employed to further refine the modality-specific features. Lastly, the segmentation head processes the fused features to generate the final prediction results.

introduces a language-guided semantic correlation distillation module to model both inter-modal and intra-modal semantics in the embedding space. While these methods demonstrate strong performance, they are limited by fixed input modalities during training, which restricts their adaptability to arbitrary sensor combinations. In contrast, our work proposes a general arbitrary-modal semantic segmentation framework capable of handling diverse sensor modalities within a generalist model.

**Vision language model.** Vision-language understanding has seen significant progress through scalable pre-training strategies. CLIP Radford et al. (2021) introduced contrastive image-text alignment, laying the foundation for multi-modal representation learning. ALIGN Jia et al. (2021) further improved robustness by leveraging noisy supervision from large-scale web data. BLIP Li et al. (2022) unified vision-language understanding and generation through bootstrapped pre-training. Recent advances in prompting techniques Zhou et al. (2022b) and lightweight adapters such as LLaMA-Adapter Zhang et al. (2024) have enhanced task adaptation efficiency. DA-CLIP Luo et al. (2024) proposed a novel dual-encoder framework in which an image controller—initially a duplicate of CLIP's image encoder—learns to control the original encoder and generate degeneration embeddings. Despite their remarkable capabilities, these vision-language models are primarily trained on natural image-text pairs and are not inherently designed for tasks involving RGB-X modalities, posing challenges in adapting them to multi-modal semantic segmentation. To address this gap, our work adopts LoRA-based parameter-efficient fine-tuning Hu et al. (2022) to adapt CLIP for cross-modal alignment. This approach enables the integration of vision-language pretraining with diverse sensor modalities, thereby enhancing the model's ability for comprehensive scene understanding.

## 3 METHODOLOGY

In this section, we present SegRGB-X, a general RGB-X semantic segmentation model. We begin with an overview of the overall pipeline in Sec. 3.1, followed by detailed descriptions of the three key components: MA-CLIP in Sec. 3.2, modality-aligned embedding in Sec. 3.3, and the DSRM in Sec. 3.4. Loss function is introduced in Sec. 3.5.

## 3.1 OVERVIEW

As illustrated in Fig. 2, our proposed SegRGB-X model comprises a backbone with modality-aligned embeddings and a DSRM, an MA-CLIP, and a segmentation head. MA-CLIP is first pre-trained to extract modality-specific representations from diverse input sources and is then frozen to serve as a modality information provider. The RGB images and arbitrary modality inputs are processed in parallel through MA-CLIP to generate feature embeddings ($S^r$ and $S^m$). The backbone consists of four stages. In each stage, shared-weight Transformer blocks Xie et al. (2021) are employed to process two modality-aligned embeddings simultaneously. The Transformer blocks share weights across modalities to ensure consistent and efficient feature extraction. During feature fusion, the FRM and FFM modules Zhang et al. (2023b) are used to integrate features from the RGB and complementary modality branches. In the final stage, a DSRM is introduced to adaptively refine the modality-specific features. Finally, the segmentation head processes these fused features from 4 stages to produce the final semantic segmentation predictions.

## 3.2 MODALITY-AWARE CLIP

The objective of Modality-aware CLIP (MA-CLIP) is to enable a pre-trained CLIP model to extract modality-specific representations from diverse input modalities. As illustrated in Fig. 3, MA-CLIP freezes both the text and image encoders of the original CLIP architecture Radford et al. (2021). To adapt the image encoder for different modalities, we introduce a LoRA pool that represents the set of supported modalities across all datasets. Each LoRA module Hu et al. (2022) in the pool is trained using the contrastive loss Radford et al. (2021), with gradients flowing through both the image encoder and the corresponding LoRA module. In the text branch, RGB images are first processed by the LLaMA-Adapter Zhang et al. (2024) to generate high-quality text captions ($T$), which are then passed through the text encoder to produce text embeddings ($S^t$). The RGB images ($X^{rgb}$) are concatenated with arbitrary modality inputs ($X^m$) to form an in-

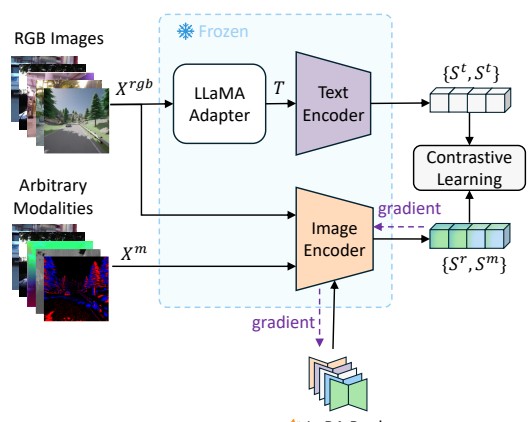

Figure 3: Structure of our MA-CLIP. MA-CLIP enhances the standard CLIP architecture Radford et al. (2021) by incorporating modality-aware cross-modal learning between textual and visual representations.

put pair ($\{X^{rgb}, X^m\}$). For each pair, the corresponding LoRA module is selected from the pool and additively integrated into the image encoder, producing an adapted encoder that outputs both RGB feature embeddings ($S^r$) and modality embeddings ($S^m$). For contrastive learning, the text embeddings ($S^t$) are repeated once.

**Optimizing the MA-CLIP.** The optimizing process keeps the weights of the pre-trained CLIP architecture frozen while exclusively optimizing the LoRA modules. To enhance the distinctiveness of the multi-modal embedding spaces, we employ the contrastive loss Radford et al. (2021) across multiple modalities, which can be formalized as:

$$\mathcal{L} = \mathcal{L}_{\text{contrastive}}(\{S^t, S^t\}, \{S^r, S^m\}). \tag{1}$$

## 3.3 MODALITY-ALIGNED EMBEDDING

As shown in Fig. 2, feature embeddings ($S^r$ and $S^m$) generated by MA-CLIP are transformed by MLP layers into stage-specific control prompts ($C^r_{i\in[1,4]}$ and $C^m_{i\in[1,4]}$), which are then combined with the input embeddings ($E^r_{i\in[1,4]}$ and $E^m_{i\in[1,4]}$) and modality-aligned prompts ($P^r_{i\in[1,4]}$ and $P^m_{i\in[1,4]}$) before being fed into the Transformer blocks. Notably, these modality-aligned prompts are proposed to bridge the feature gap between the input embeddings and control prompts, facilitating improved cross-modal alignment.

### 3.4 DOMAIN-SPECIFIC REFINEMENT MODULE

As shown in Fig. 4, the shared Domain-Specific Refinement Module (DSRM) is designed to refine modality-specific features. It operates on modality pairs $(F_4^r, S^r)$ and $(F_4^m, S^m)$, with a total of four DSRM modules integrated throughout the network. The DSRM adopts a query-based prompt architecture to enhance feature representations. Specifically, the input features $F$ first pass through a Global Average Pooling (GAP) layer to capture global channel-wise information, followed by an MLP for feature transformation. The transformed features are then normalized via a softmax operation. To model complex feature correlations, we introduce a learnable universal prompt $U$. A dot product is performed between the transformed features and the universal prompt to generate the query vector $Q_c$ for the subsequent channel attention, facilitating effective intra-modal interactions. Concurrently, the original input features $F$ are used to generate the corresponding key

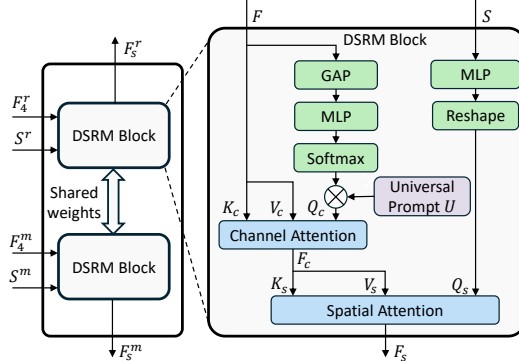

Figure 4: Domain-Specific Refinement Module (DSRM). It contains two identical DSRM blocks with shared weights, each processing a different modality pair $(F_4^r, S^r)$ and $(F_4^m, S^m)$ to produce enhanced features ($F_s^r$ and $F_s^m$).

$(K_c)$ and value $(V_c)$ vectors for the attention operation. The computation can be formulated as:

$$Q_c = W_c^Q \cdot (\text{Softmax}(\text{MLP}(\text{GAP}(F))) \cdot U), K_c = W_c^K \cdot F, V_c = W_c^V \cdot F, \quad (2)$$

$$F_c = \text{MHSA}(Q_c, K_c, V_c), \quad Q_c, K_c, V_c, F_c \in B \times C \times N. \quad (3)$$

The outputs ($F_c$) are further transformed to generate the key ($K_s$) and value ($V_s$) vectors for the spatial attention mechanism. For query vector ($Q_s$), the modality-specific features ($S$) are first transformed through an MLP layer to align with the required dimensionality. Multi-Head Cross-Attention (MHCA) is then applied to compute the final feature representations, effectively capturing both spatial dependencies and modality-specific interactions. The process can be represented as:

$$Q_s = W_s^Q \cdot \text{Reshape}(\text{MLP}(S)), K_s = W_s^K \cdot F_c, V_s = W_s^V \cdot F_c, \quad (4)$$

$$F_s = \text{MHCA}(Q_s, K_s, V_s), \quad Q_s, K_s, V_s, F_s \in B \times N \times C. \quad (5)$$

### 3.5 LOSS FUNCTION

To enable joint training across multiple datasets, we construct a universal dataset that unifies all semantic labels from the involved modality-specific datasets. Moreover, we extend the standard cross-entropy loss by incorporating dataset-specific loss terms, ensuring effective learning across heterogeneous label distributions. The overall loss function is defined as follows:

$$\mathcal{L} = \mathcal{L}_{ce}(X_i, Y_i) + \mathcal{L}_{ce}(\text{remap}(X_i), \text{remap}(Y_i)). \quad (6)$$

where remap$(\cdot)$ denotes the operation that maps the unified label space back to the original label space of each dataset.

## 4 EXPERIMENTS

### 4.1 EXPERIMENTAL SETTINGS

**Datasets.** To train our model across multiple datasets simultaneously, we construct a joint dataset by standardizing the label spaces of 5 benchmark datasets: DeLiVER Zhang et al. (2023b), MFNet Ha et al. (2017), NYUDepthV2 Silberman et al. (2012), RGB-P Xiang et al. (2021), and UrbanLF Sheng et al. (2022). DeLiVER Zhang et al. (2023b) is a synthetic autonomous driving dataset generated using the CARLA simulator Dosovitskiy et al. (2017). It contains RGB, depth, Li-DAR, and event modalities, with 7,885 front-view samples. The data is split into 3,983/2,005/1,897 for training, validation, and testing, respectively, at a resolution of $1042 \times 1042$, and includes 25 semantic classes. It simulates four adverse weather conditions (cloudy, foggy, night, and rainy) and

Table 1: Quantitative comparison of segmentation performance on five multi-modal datasets.

| Method | Model | Mean (%) | DeLiVER (E) | MFNet (T) | NYU (D) | RGB-P (P) | UrbanLF (LF) |
|---|---|---|---|---|---|---|---|
| CMX Zhang et al. (2023a) | Specialist Model | 63.19 | 51.77 | 54.54 | 48.49 | 82.09 | 79.06 |
| CMNeXt Zhang et al. (2023b) | | 63.56 | 51.78 | 55.35 | 48.75 | 82.28 | 79.65 |
| Gemini Fusion Jia et al. (2024b) | | 63.75 | 51.24 | 53.73 | 52.18 | 82.19 | 79.43 |
| Stitch Fusion Li et al. (2024) | | 64.51 | 51.81 | 55.90 | **52.64** | 82.47 | 79.71 |
| Ours | Generalist Model | **65.03** | **51.83** | **56.93** | 47.77 | **87.39** | **81.21** |

five types of sensor degradation (motion blur, overexposure, underexposure, LiDAR jitter, and event low resolution). MFNet Ha et al. (2017) is an urban street dataset comprising 1,569 RGB-thermal image pairs captured at a resolution of $640 \times 480$, annotated with 8 semantic classes. The dataset is evenly divided between daytime (820 pairs) and nighttime (749 pairs) conditions and split into 50%/25%/25% for training, validation, and testing. NYUDepthV2 Silberman et al. (2012) is an indoor RGB-D semantic segmentation dataset with 1,449 image pairs, divided into 795 training and 654 testing samples at a resolution of $640 \times 480$. The dataset provides annotations for 40 semantic classes. RGB-P Xiang et al. (2021) (ZJU RGB-Polarization) comprises 394 RGB-polarization image pairs collected from urban driving scenes. Each sample includes an RGB image and polarization information synthesized from four images captured at polarization angles of $0°$, $45°$, $90°$, and $135°$. UrbanLF Sheng et al. (2022) is a large-scale light field semantic segmentation dataset containing 1,074 samples, split into 824 real-world and 250 synthetic scenes. Each light field sample includes 81 views. Real-world samples are captured at a resolution of $623 \times 432$, while synthetic samples, rendered using Blender, are at $640 \times 480$ resolution.

**Implementation details.** We train our models using $2\times$ NVIDIA A5000 GPUs with a batch size of 8. The initial learning rate is set to $1 \times 10^{-4}$ and is scheduled using the poly learning rate policy with a power of 0.9 over 200 epochs. Input images are uniformly augmented across all datasets using the following strategies: random resizing with a scale ratio in the range of [0.5, 2.0], random horizontal flipping, random color jittering, random Gaussian blur, and random cropping to a fixed resolution of $512 \times 512$ to achieve batched training of various image shapes from different datasets. To initialize the model backbone, we load ImageNet-1K Deng et al. (2009) pre-trained weights.

## 4.2 QUANTITATIVE ANALYSIS

We conduct comprehensive experiments on five multi-modal semantic segmentation datasets to evaluate the performance of our proposed SegRGB-X model. Tab. 1 presents a comparison between our general SegRGB-X model and several specialist models. Specifically, the DeLiVER Zhang et al. (2023b) dataset provides RGB and event (**E**) modalities; the MFNet Ha et al. (2017) and RGB-P Xiang et al. (2021) datasets offer RGB with thermal (**T**) and polarization (**P**) modalities, respectively; the NYUDepthV2 Silberman et al. (2012) dataset contains RGB and depth (**D**) data; and the UrbanLF Sheng et al. (2022) dataset includes sub-aperture light field (**LF**) images, from which we use the first sub-aperture image as the complementary modality.

In this comparison, specialist models are trained individually on each modality dataset, while our generalist model is trained once across all modality datasets and can perform inference with any given modality input. For a fair comparison, baseline methods including CMX Zhang et al. (2023a), CMNeXt Zhang et al. (2023b), Gemini Fusion Jia et al. (2024a), and Stitch Fusion Li et al. (2024) are re-implemented using their official open-sourced codes and trained under the same experimental settings.

Our general SegRGB-X model achieves the best overall performance, reaching an average mIoU of **65.03%** across the five datasets. Moreover, the proposed model outperforms the second best method with gains of **+1.03%** on MFNet, **+4.92%** on RGB-P, and **+1.5%** on UrbanLF. However, the model achieves a moderate result of 47.77% on NYUDepthV2. We attribute this to a domain distribution gap: NYUDepthV2 is the only indoor dataset, while the other four datasets focus on outdoor driving scenes, leading to challenges in achieving cross-scenario generalization.

**Fine-tuning.** Our generalist model, SegRGB-X, demonstrates strong performance through joint training on five diverse multi-modal semantic segmentation datasets. To further enhance segmenta-

Table 2: Fine-tuning comparison of segmentation performance on five multi-modal datasets.

| Method | Mean (%) | DeLiVER (E) | MFNet (T) | NYU (D) | RGB-P (P) | UrbanLF (LF) |
|---|---|---|---|---|---|---|
| SegRGB-X | 65.03 | 51.83 | 56.93 | 47.77 | 87.39 | 81.21 |
| SegRGB-X (Fine-tuned) | **65.72** | **52.54** | **57.28** | **49.02** | **88.36** | **81.65** |

tion accuracy, we fine-tune the pretrained SegRGB-X model on each individual dataset. As shown in Tab. 2, this dataset-specific fine-tuning leads to consistent performance gains, resulting in an overall mean mIoU increase of +0.69%. Specifically, we observe gains of +0.71% on DeLiVER Zhang et al. (2023b), +0.35% on MFNet Ha et al. (2017), +1.25% on NYUDepthV2 Silberman et al. (2012), +0.97% on ZJU RGB-P Xiang et al. (2021), and +0.44% on UrbanLF Sheng et al. (2022). These results confirm that while SegRGB-X is effective as a generalist model, additional fine-tuning can further optimize its performance for specific domains.

**Efficiency analysis.** We further compare the efficiency of our method against existing SOTA multi-modal semantic segmentation models in terms of running time. As presented in Tab. 3, our approach achieves a favorable balance between accuracy and efficiency. Among specialist models, CMNeXt Zhang et al. (2023b) attains the lowest latency of 16.64 ms, whereas Gemini Fusion Jia et al. (2024a) is the most time-consuming, with a latency of 24.37 ms.

Table 3: Efficiency analysis of different methods.

| Method | Mean (%, ↑) | Time (ms, ↓) |
|---|---|---|
| CMX Zhang et al. (2023a) | 63.19 | 17.05 |
| CMNeXt Zhang et al. (2023b) | 63.56 | **16.64** |
| Gemini Fusion Jia et al. (2024a) | 63.75 | 24.37 |
| Stitch Fusion Li et al. (2024) | 64.51 | 17.88 |
| Ours | **65.03** | 31.51 |

However, these specialist models require separate deployments tailored to each specific modality pair, limiting scalability. In contrast, our method achieves generalized multi-dataset semantic segmentation with a unified architecture while maintaining a comparable latency of 31.51 ms. This demonstrates the practicality and deployability of our method in real-world applications, particularly in resource-constrained environments where efficiency and flexibility across diverse sensor modalities are critical.

### 4.3 QUALITATIVE ANALYSIS

We present visualizations of semantic segmentation results, t-SNE plots of modality embeddings produced by MA-CLIP, and multi-stage feature maps from the backbone network. All selected samples are from unseen data during training, demonstrating the strong generalization capability of our SegRGB-X model.

**t-SNE analysis.** As illustrated in Fig. 5, we present a t-SNE visualization of the modality embeddings generated by our proposed MA-CLIP across five modalities: event, thermal, depth, polarization, and light field. The visualization shows that embeddings from each modality form well-separated clusters with clear boundaries and no overlap, indicating MA-CLIP's strong capability for modality-specific feature extraction. Additionally, the distances between cluster centers reflect the relative feature similarity among modalities. These results demonstrate that MA-CLIP effectively produces discriminative modality embeddings even on unseen inputs, validating its effectiveness across diverse modalities.

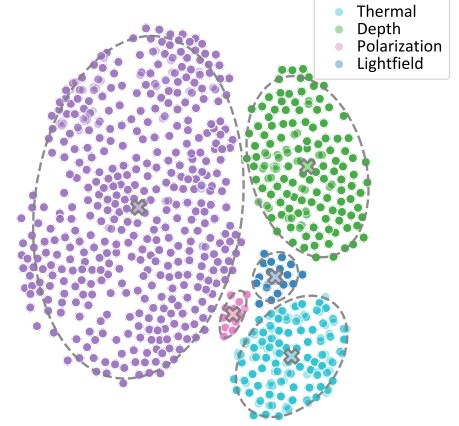

Figure 5: t-SNE visualization of modality embeddings from MA-CLIP. Each cluster corresponds to a specific modality, showing clear separation and highlighting the model's strong ability to extract distinctive modality-specific representations.

**Segmentation predictions.** As shown in Fig. 6, we present qualitative comparisons of semantic segmentation results produced by CMNeXt Zhang et al. (2023b), Gemini Fusion Jia et al. (2024a),

Table 4: Ablation studies on single-modality and joint training.

| Method | Model | Mean (%) | DeLiVER (E) | MFNet (T) | NYU (D) | RGB-P (P) | UrbanLF (LF) |
|---|---|---|---|---|---|---|---|
| Stitch Fusion Li et al. (2024) | Specialist Model | 64.51 | 51.81 | 55.90 | **52.64** | 82.47 | 79.71 |
| Ours | | 64.38 | 51.83 | 56.93 | 47.77 | 87.39 | 81.21 |
| Ours | Generalist Model | **65.03** | **51.83** | **56.93** | 47.77 | **87.39** | **81.21** |

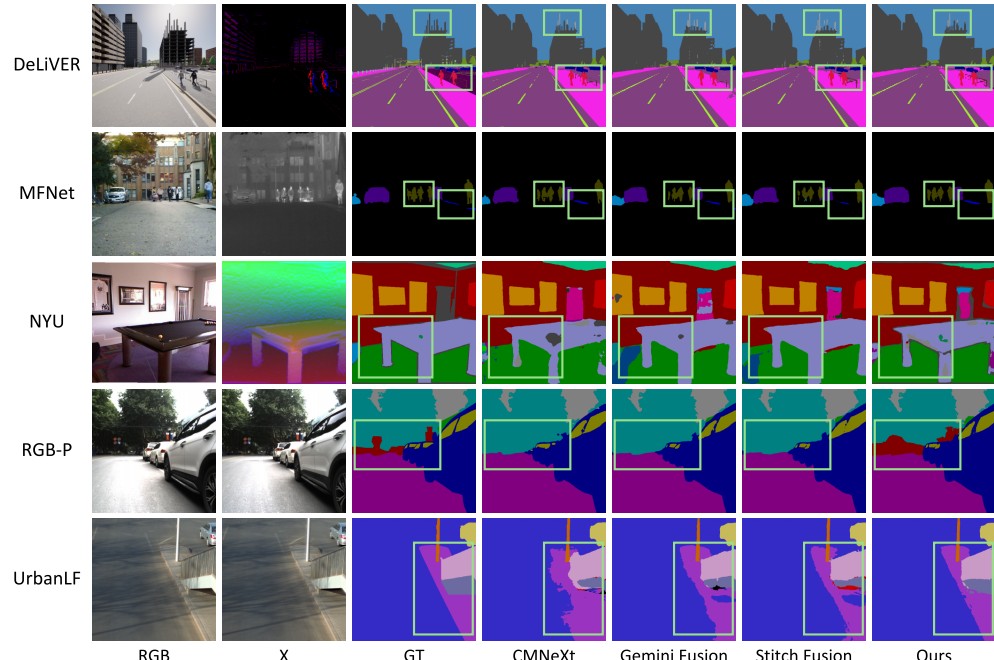

Figure 6: Comparison of segmentation predictions from CMNeXt Zhang et al. (2023b), Gemini Fusion Jia et al. (2024a), Stitch Fusion Li et al. (2024), and our proposed method across all five datasets using unseen input samples.

Stitch Fusion Li et al. (2024), and our proposed SegRGB-X model across five multi-modal datasets. Compared to the other methods, our model yields more accurate and coherent segmentation results. For instance, SegRGB-X successfully identifies a building occluded by trees in the RGB-P input, whereas other models produce incorrect predictions. Similarly, our method generates more reasonable segmentation outputs on the UrbanLF dataset. Comparable improvements can be observed across the remaining datasets. These results demonstrate the strong generalization capability of our generalist model, which is even better than specialist models trained for each individual dataset.

## 4.4 ABLATION STUDIES

**Single-modality ablation studies.** We transform the general model into a specialist one that focuses on each single modality to validate the effectiveness of joint training. As shown in Tab. 4, our model achieves the overall performance of 64.38% with better mIOU on the MFNet, RGB-P, and UrBanLF datasets in the single modality ablation results compared with Stitch Fusion Li et al. (2024), even though we mainly focus on the generalization of different modalities and datasets. The joint training further enhances the overall mean IOU to 65.03% (+0.65%), indicating the effectiveness of the joint training strategy.

Table 5: Ablation studies on key components of the proposed SegRGB-X model.

| MA-CLIP | Modality-aligned Prompts | DSRM | Mean (%) |
|---|---|---|---|
| ✗ | ✗ | ✗ | 62.17 |
| ✗ | ✓ | ✓ | 64.33 |
| ✓ | ✗ | ✓ | 64.43 |
| ✓ | ✓ | ✗ | 64.55 |
| ✓ | ✓ | ✓ | **65.03** |

**Impact of key components.** We conduct experiments to evaluate the effectiveness of the key components in our SegRGB-X model, including MA-CLIP, modality-aligned prompts, and the DSRM.

The experimental results, summarized in Tab. 5, are obtained by incrementally disabling each component and measuring the average performance across the five multi-modal datasets. Removing MA-CLIP results in a performance drop of $0.70\%$, indicating the importance of cross-modal pre-training for modality-specific feature extraction. Excluding modality-aligned prompts leads to a $0.6\%$ decrease, demonstrating their role in bridging the feature gap between input embeddings and control prompts. Omitting DSRM yields a $0.48\%$ performance degradation, underscoring the value of semantic refinement in the final stage. Notably, when all three components are removed, the performance drops significantly by $2.86\%$. These results validate the contribution of each individual component and demonstrate the effectiveness of our designs.

**The impact of different modality-aligned embedding mechanisms.** To investigate the effectiveness of different modality-aligned embedding mechanisms, we evaluate several architectural variants by altering the pairing strategy between the input embeddings and the control embeddings produced by MA-CLIP. The experimental results are presented in Tab. 6. In particular, the "Aligned" configuration pairs input RGB embeddings with RGB embeddings from MA-CLIP and input modality embeddings with the corresponding modality embeddings from MA-CLIP. In contrast, the "Cross-modal" setting aligns RGB input embeddings with modality embeddings from MA-CLIP. Finally, the "RGB-dominant" approach pairs both the input RGB and modality embeddings with RGB embeddings from MA-CLIP, under the hypothesis that RGB features provide more robust semantic priors. The results show that both the "Aligned" and "Cross-modal" strategies yield suboptimal performance. In comparison, the "RGB-dominant" configuration achieves the highest mean mIoU of 65.03%, demonstrating the effectiveness of using RGB-guided semantic representations as a unified control prior for diverse modalities in multi-modal segmentation tasks.

Table 6: The performance of different modality-aligned embedding mechanisms.

| Configuration | Mean (%) |
|---|---|
| Aligned | 64.18 |
| Cross-modal | 64.12 |
| RGB-dominant | **65.03** |

**The influence of different modality pairs in DSRM.** We conduct experiments to investigate the impact of different modality pair configurations within the DSRM, while maintaining the "RGB-dominant" strategy in the modality-aligned embedding mechanism. As shown in Tab. 7, the "Aligned" configuration—where each input modality in DSRM is paired with the corresponding modality-specific embedding from MA-CLIP—achieves the best performance with a mean mIoU of 65.03%. This result highlights the effectiveness of using modality-consistent semantic cues from MA-CLIP to enhance feature refinement. We also explore a "Cross-modal" configuration that pairs each input with embeddings from a different modality, as well as a "RGB-dominant" configuration where all inputs are paired with RGB embeddings from MA-CLIP. However, these alternative strategies result in reduced performance, with mean mIoUs of 64.05% and 64.32%, respectively. These findings underscore the importance of modality-specific alignment in DSRM to fully exploit the semantic information encoded in multi-modal inputs.

Table 7: The performance of different modality pairs in DSRM.

| Configuration | Mean (%) |
|---|---|
| Aligned | **65.03** |
| Cross-modal | 64.05 |
| RGB-dominant | 64.32 |

## 5 CONCLUSION

In this work, we propose SegRGB-X, a general RGB-X semantic segmentation model designed to handle diverse input modalities. The proposed model incorporates three key components: an MA-CLIP, a modality-aligned embedding mechanism, and a DSRM. Extensive experiments conducted on five multi-modal segmentation datasets demonstrate that our approach achieves excellent performance both quantitatively and qualitatively.

**Limitations.** We have validated the effectiveness of the proposed method on five modalities: RGB, event, thermal, depth, polarization, and light field. Extending the current framework to accommodate additional modalities remains a promising direction for future research.

ETHICS STATEMENT

Our general SegRGB-X multi-modal semantic segmentation framework trains a single unified model across multiple datasets, enabling robust adaptation to diverse sensor modalities and environments. This approach achieves SOTA accuracy and reliability, which are essential for safety-critical applications such as autonomous driving and robots. By consolidating multiple modality-specific models into one, SegRGB-X reduces redundant model storage and consequently lowers the carbon footprint associated with AI deployment. Moreover, its superior effectiveness and robustness enhance real-world safety under a wide range of conditions. We recognize the dual-use nature of this technology. While its intended applications aim to advance societal benefits, there exist ethical concerns regarding potential misuse—such as the displacement of workers due to increased automation in autonomous driving. We explicitly oppose harmful exploitation and strongly advocate for the establishment of strict governance frameworks to ensure responsible development and deployment, minimizing adverse societal impacts.

REPRODUCIBILITY STATEMENT

The experiments are run with public available datasets and backbones with a fixed random seed. We will release the full code, configurations, preprocessing and evaluation scripts and our trained weights once upon acceptance.

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

# A  APPENDIX

## A.1  USE OF LLMS

We used LLMs only for grammar, wording, and formatting edits. All technical content, analyses, and reported results were authored and verified by the authors. There is no scientific claims or data that were generated by the LLMs.

## A.2  THEORETICAL ANALYSIS OF KEY COMPONENTS

**MA-CLIP.** At the core of MA-CLIP is the goal of controlling a pre-trained CLIP Radford et al. (2021) model to output the coarse-grained semantic embeddings via contrastive learning. Specifically, we leverage LoRA Hu et al. (2022) to overcome the insufficient primary knowledge of CLIP Radford et al. (2021) for the supplementary modalities. We freeze all weights of the pre-trained CLIP Radford et al. (2021) model and only fine-tune the LoRA Hu et al. (2022) modules. Since our training dataset is tiny compared to the web-scale datasets used in VLMs, this LoRA strategy alleviates overfitting while preserving the capability of the original image encoder. The embeddings are further integrated into the segmentation backbone to assist the fine-grained semantic segmentation task.

**Modality-aligned embeddings.** Prompts are commonly used to give hints for model to adjust its behavior. Following this idea, we have input patch embeddings $E \in \mathbb{R}^{N \times D}$ and some modality-specific prompts as embeddings $M \in \mathbb{R}^{K \times D}$. The input then becomes $\hat{E} = [E; M] \in \mathbb{R}^{(N+K) \times D}$. To perform attention, the projection matrices are first applied: $Q = \hat{E}W_Q, K = \hat{E}W_K, V = \hat{E}W_V$. Then the attention matrix can be decoupled to: $\boldsymbol{A} = \text{softmax}\left(\frac{QK^T}{\sqrt{D_h}}\right) = \begin{bmatrix} \boldsymbol{A}_{EE} & \boldsymbol{A}_{EM} \\ \boldsymbol{A}_{ME} & \boldsymbol{A}_{MM} \end{bmatrix}$, where $\boldsymbol{A}_{EE}$ and $\boldsymbol{A}_{MM}$ describe the self-attention between patch embeddings and modality

prompts, while $\boldsymbol{A}_{EM}$ and $\boldsymbol{A}_{ME}$ describes the interaction between patch embeddings and modality prompts. The output can be formulated as $\boldsymbol{O} = \boldsymbol{A}V = \begin{bmatrix} \boldsymbol{O}_E \\ \boldsymbol{O}_M \end{bmatrix} = \begin{bmatrix} \boldsymbol{A}_{EE}V_E & \boldsymbol{A}_{EM}V_M \\ \boldsymbol{A}_{ME}V_E & \boldsymbol{A}_{MM}V_M \end{bmatrix}$. To prevent the propagation of error information between different encoder layers, the $\boldsymbol{O}_M$ is deprecated and only $\boldsymbol{O}_E$ is retained. The $i$-th output patch of $\boldsymbol{O}_E$ can be formulated as

$$\underbrace{\sum_{j=1}^{N} \boldsymbol{A}_{EE}(i,j)\boldsymbol{v}_j}_{\text{self-attention between patches}} + \underbrace{\sum_{k=1}^{K} \boldsymbol{A}_{EM}(i,k)\boldsymbol{v}_{m_k}}_{\text{prompts specific adaption}} \text{ with the original self-attention and additional prompts}$$

specific adaption as marked in the formulation. In practice, we split the modality prompts $M$ into stage-specific control prompts $C$, generated by MA-CLIP using the RGB images, and learnable modality-aligned prompts $P$. The modality-aligned prompts are responsible for modality information and local details that are potentially missed by the global stage-specific control prompts, serving as supplementary information. The choice of leveraging RGB images for prompts stems from their semantically rich property. We conducted ablation experiments for the integration types of embeddings generated by MA-CLIP in Table 1 of our supplementary material, and the results validate the theoretical statements.

**DSRM.** Modalities are usually not independent of each other but correlated. For instance, light flow and polarization modalities are more RGB-like, while depth and thermal modalities share the same image structure as RGB images, with the contents from another point of view. And the event modality only shows partial structure when the brightness changes. Thus, we can conclude that correlated modalities should share similar features with each other. Thus, suppose we have a collection of $K$ feature vectors $U \in \mathbb{R}^{K \times D}$, and for each modality m, there exists a binary indicator $\chi \in \{0,1\}^K$ for the existence of a certain feature; then the feature for modality $m$ is $\chi \cdot U$, aiming to select the feature where $\chi = 1$ and discard where $\chi = 0$. Through the channel attention, the input features $F$ are reweighted by the cosine similarity with $\chi \cdot U$. In practice, we loosen the binary restriction and extend it to a percentage for flexibility. Furthermore, we design a simple network with a global average pooling, MLP, and softmax operation to learn the indicator from input features. Similarly, the output of the channel attention will be reweighted by the cosine similarity with the modality embeddings $S$ in the spatial attention.

## A.3 More Quantitative Analysis

**Influence of different DSRM structures.** To further assess the effectiveness of the proposed DSRM, we explore various structure designs, as illustrated in Fig. 7. The results are summarized in Tab. 8. Structures $(a)$–$(d)$ utilize the input features as queries and the prompts as keys and values in the first attention module but consistently perform suboptimally. Structure $(e)$, which applies spatial attention in both modules using prompts as queries and inputs as keys and values, shows a performance decline to 62.90% accuracy, despite having the highest model complexity in terms of parameters and GFLOPs. In contrast, structure $(h)$—which sequentially applies channel attention followed

Table 8: Influence of different structure designs for DSRM. (a)-(i) are listed in Fig. 7.

| Structure | Params (M) | GFLOPs | Mean (%)) |
|---|---|---|---|
| $(a)$ | 176.70 | 111.15 | 64.68 |
| $(b)$ | 143.63 | 99.74 | 64.83 |
| $(c)$ | 160.96 | 105.65 | 64.68 |
| $(d)$ | 159.38 | 105.24 | 64.79 |
| $(e)$ | 230.61 | 113.03 | 62.90 |
| $(f)$ | 203.31 | 106.58 | 64.74 |
| $(g)$ | 212.53 | 109.79 | 64.59 |
| $(h)$ | 159.40 | 108.31 | **65.03** |
| $(i)$ | 156.32 | 107.01 | 47.58 |

by spatial attention using prompts as queries—achieves the best result, attaining 65.03% accuracy while maintaining computational efficiency. Notably, reversing the attention order (spatial followed by channel attention in structure $(g)$) leads to a 0.44% drop in accuracy.

To examine the role of our universal prompt ($U$), we replace it with the input feature ($F$) in structure $(i)$. This substitution results in a significant performance degradation of 17.45% (from 65.03% to 47.58%), underscoring the importance of the learnable universal prompt in capturing intra-modality feature correlations.

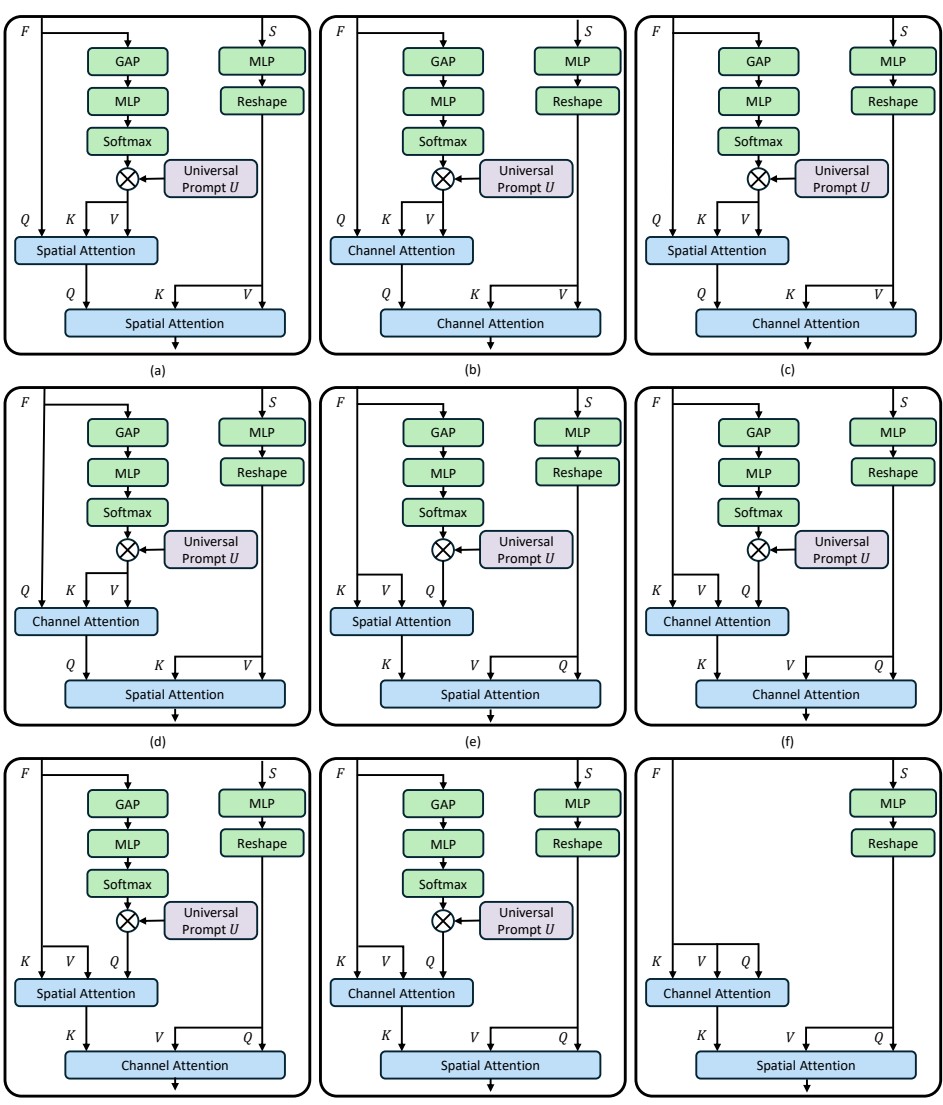

Figure 7: Structures (a)–(d) adopt a key-value-based prompt design, while (e)–(i) follow a query-based prompt formulation. Each structure explores four combinations of spatial and channel attention mechanisms.

### A.4 MORE QUALITATIVE ANALYSIS

**Feature map visualization.** In Fig. 8, we visualize the feature maps of RGB and the corresponding complementary modality across the four stages of the backbone network. All feature maps are resized to the same resolution. Features from earlier stages primarily capture low-level local patterns such as edges and corners, while those from deeper stages focus on higher-level semantics. These visualizations demonstrate that our approach effectively extracts modality-specific features across diverse modality domains by leveraging MA-CLIP and modality-aligned embeddings. Notably, the feature maps from different modalities exhibit complementary characteristics. Furthermore, compared to Stage 3, the feature maps from Stage 4 show enhanced focus on key semantic regions, attributed to the refinement effect of the proposed DSRM. For instance, building structures in the event modality of the DeLiVER dataset and the chair in the depth modality of the NYUDepthV2 dataset are more distinctly highlighted. These results validate the effectiveness of DSRM in refining modality-specific representations.

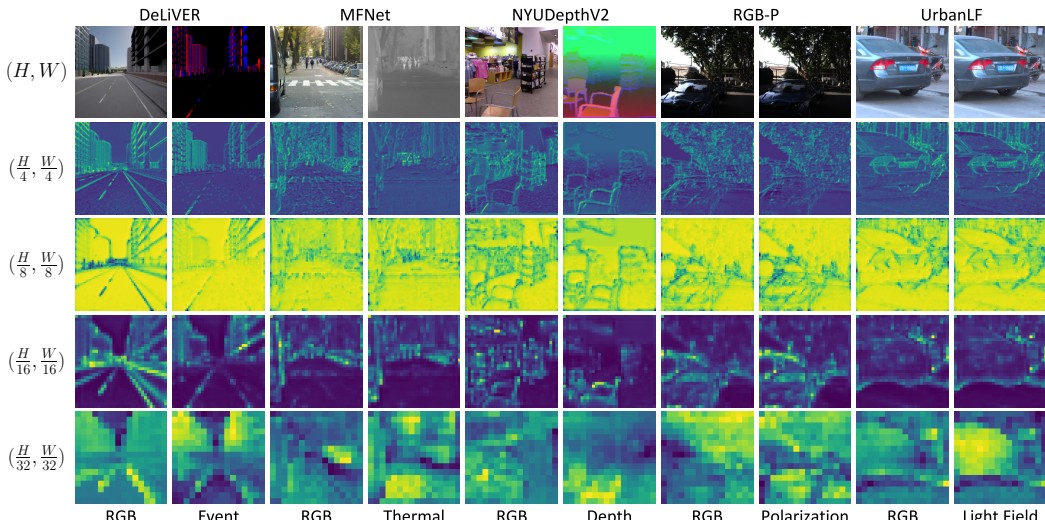

Figure 8: Feature map visualizations of multi-modal features extracted from the four stages of the backbone network across all five datasets. The column corresponds to a specific modality and its associated dataset, illustrating the hierarchical feature representations learned at different scales.

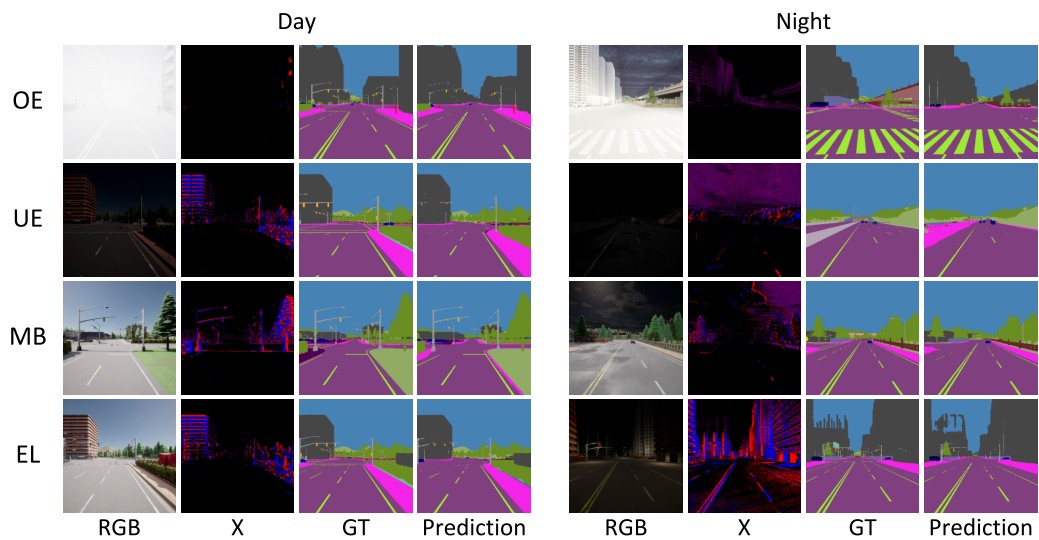

Figure 9: Visualization of segmentation predictions on challenging edge cases from the DeLiVER Zhang et al. (2023b) dataset. The examples include various adverse conditions: **OE** (Over-Exposure), **UE** (Under-Exposure), **MB** (Motion Blur), and **EL** (Event Low-resolution). Our model, SegRGB-X, demonstrates robust segmentation performance under these extreme conditions, showcasing its generalization ability across diverse visual degradations.

**Segmentation predictions in edge cases.** As shown in Fig. 9, we visualize the segmentation predictions generated by our proposed SegRGB-X model under various challenging edge-case scenarios, including over-exposure, under-exposure, motion blur, and event low resolution, across both day and night conditions. The results illustrate that our model maintains high-quality segmentation performance despite environmental perturbations and sensor degradation. This demonstrates the robustness and strong generalization capability of SegRGB-X in handling adverse visual conditions commonly encountered in real-world applications.

**Feature map visualization in edge cases.** In Fig. 10, we visualize the feature maps of RGB and event inputs across the four stages of the backbone network under various challenging edge cases—including over-exposure, under-exposure, motion blur, and event low-resolution—captured

Day                                Night

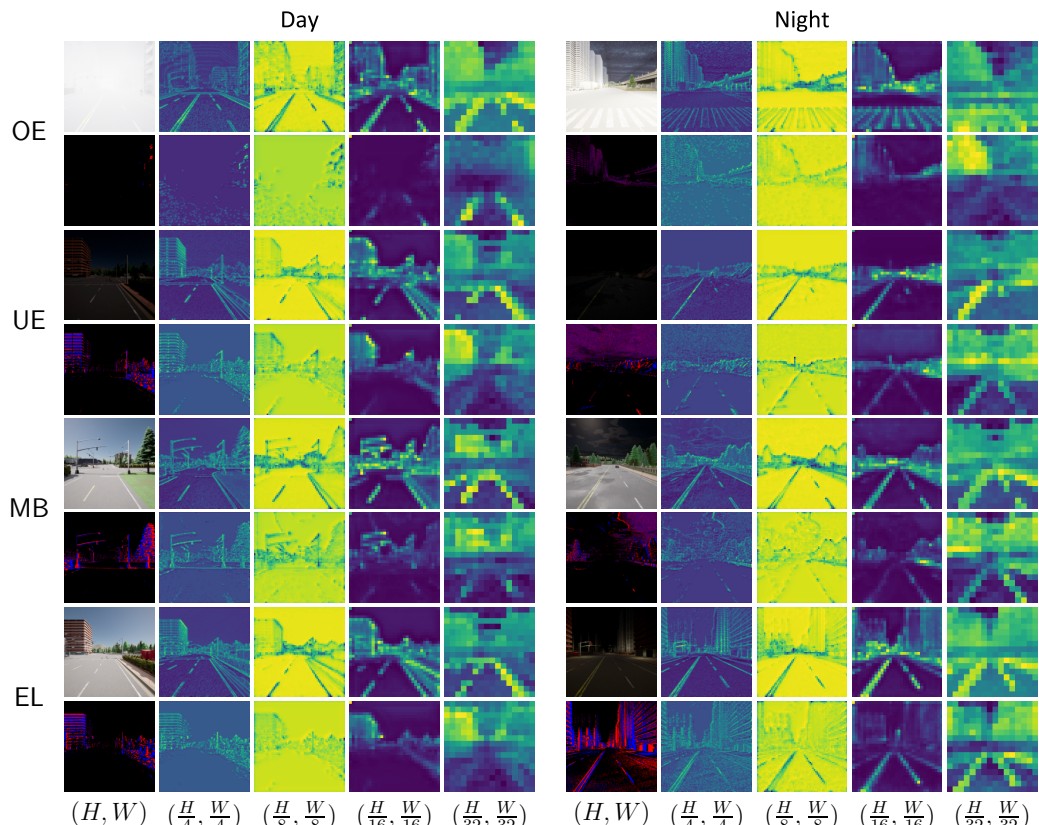

OE

UE

MB

EL

$(H, W)$ $\left(\frac{H}{4}, \frac{W}{4}\right)$ $\left(\frac{H}{8}, \frac{W}{8}\right)$ $\left(\frac{H}{16}, \frac{W}{16}\right)$ $\left(\frac{H}{32}, \frac{W}{32}\right)$     $(H, W)$ $\left(\frac{H}{4}, \frac{W}{4}\right)$ $\left(\frac{H}{8}, \frac{W}{8}\right)$ $\left(\frac{H}{16}, \frac{W}{16}\right)$ $\left(\frac{H}{32}, \frac{W}{32}\right)$

Figure 10: Feature map visualizations of multi-modal features extracted from the four stages of the backbone network in challenging edge-case scenarios of the DeLiVER Zhang et al. (2023b) dataset (**OE**: Over-Exposure; **UE**: Under-Exposure; **MB**: Motion Blur; **EL**: Event Low-resolution). The rows depict the hierarchical feature representations learned at different scales throughout the backbone.

in both day and night environments. The visualizations show that the feature representations remain robust despite the presence of significant sensor noise. Furthermore, the consistent performance of the DSRM module under these adverse conditions highlights its stability and effectiveness in refining modality-specific features.

## A.5 GENERALIZATION ERROR ANALYSIS

From our experimental results, we observed that our model generalized well except on the NYUDepthV2 Silberman et al. (2012) datasets. The NYUDepthV2 Silberman et al. (2012) dataset contains highly complicated indoor scenes consisting of furniture like chairs, desks, couches, etc. of different types, while the other datasets mainly contain less complicated outdoor scenes with streets, pedestrians, buildings, etc. We attribute the limited generalization on the NYUDepthV2 Silberman et al. (2012) dataset to the imbalance of input data. Given that the NYUDepthV2 Silberman et al. (2012) dataset is the only indoor dataset that occupies only 20% and the remaining 80% of the data consists exclusively of outdoor scenes, they gain priority in the data and thus weigh more in training. One potential improvement is to include more indoor datasets, such as the SUN-RGBD dataset, the Stanford2D3D dataset, and the ScanNetV2 dataset, so that both types of datasets are equally treated.

