# OpenReview forum: "SegRGB-X: General RGB-X Semantic Segmentation Model"
_ICLR.cc/2026/Conference — ICLR 2026 Conference Withdrawn Submission_

### Official Review · Reviewer_gpma · 2025-10-16

**Soundness:** 3
**Presentation:** 3
**Contribution:** 2
**Rating:** 4
**Confidence:** 5

**Summary:**

This paper proposed a universal arbitrary-modality semantic segmentation method using modality-aware CLIP, complementary learnable prompts, and a modality-aware selective adapter. The experimental results demonstrate the effectiveness of the proposed on different multimodal segmentation benchmarks.

**Strengths:**

- The proposed method introduces modality-aware CLIP to provide modality-specific scene understanding.
- Experiments are conducted on multiple benchmarks, and the results show that the proposed method can serve as a generalist model for advancing multimodal segmentation.

**Weaknesses:**

- In Table 1, the compared methods actually achieved higher results in their respective papers. For example, CMX achieved more than 92% in mIoU on RGB-P segmentation. The results on MFNet are also under-reported in this paper. This is a main concern. Please compare the methods fairly and directly report their original results.

- The proposed DSRM should be directly compared against state-of-the-art relevant modules or adapters to illustrate the superiority of the proposed module.

- Recent generalist models such as OmniSegmentor, MemorySAM, Any2Seg, etc., should be compared in the experiments to demonstrate the superiority of the proposed method.

**Questions:**

- The t-SNE visualization shows the modality embeddings from MA-CLIP, which are well separated. Would you consider showing more results or challenging cases? For example, how about the baseline model and the known CMX model? How about their modality embeddings?

- Would you consider verifying the generalization capacity of your proposed model in modality-missing, modality-degraded, or modality-incomplete scenarios? The generalist models like Any2Seg and AnySeg can also perform well in these scenarios, which could be compared.

- Would you consider presenting the performance scores in different scenarios, such as the OE, UE, MB, and EL scenarios, as in the DeLiVER benchmark? This would help enrich the analysis.

---

### Official Review · Reviewer_Lvta · 2025-10-22

**Soundness:** 2
**Presentation:** 3
**Contribution:** 2
**Rating:** 4
**Confidence:** 4

**Summary:**

Motivated by the goal of achieving state-of-the-art (SOTA) performance across diverse multimodal semantic segmentation datasets, this paper proposes a universal, arbitrary-modality semantic segmentation framework—SegRGB-X—that unifies segmentation across multiple modalities. The method introduces three key components: (1) Modality-Aware CLIP (MA-CLIP), (2) complementary learnable prompts for fine-grained feature capture, and (3) a Modality-Aware Selective Adapter (MASA). The approach is evaluated on a diverse set of datasets encompassing five complementary modalities: event, thermal, depth, polarization, and light field. The authors claim that SegRGB-X outperforms the baselines they selected.

**Strengths:**

The paper is well-motivated by the ambition to develop a single model that achieves SOTA performance across various multimodal semantic segmentation benchmarks. The proposed method trains once and performs well across all modalities—a compelling vision. The integration of multimodal CLIP features into the learning pipeline offers a novel perspective on leveraging large pre-trained vision models for downstream tasks.

**Weaknesses:**

1. The paper’s justification for its core premise is insufficiently substantiated. In the introduction, the authors hastily conclude: “Consequently, a separate model must be trained for every modality pair, leading to redundancy in both model design and training.” However, they neither theoretically analyze the commonalities and differences across modalities nor provide empirical evidence to support this claim. For instance, event camera data is extremely sparse, whereas depth provides dense geometric cues—these are fundamentally different signal types. Is a one-size-fits-all unified model truly superior to modality-specific designs? This unsubstantiated assumption undermines the foundational motivation of the work.

2. Questionable Claim of Superiority Over Specialized Models:The paper asserts that its “general-purpose” model outperforms “specialized” models, but this claim does not hold against actual SOTA methods. For example, as shown in Table 1, DFormer-v2 achieves 58.4 mIoU on NYUv2 and 67.1 mIoU on the DELIVER dataset—both significantly higher than the results reported for SegRGB-X. Even a specialized model using only RGB + Event inputs (CMNeXt-RGB-E) achieves 57.48 mIoU. These results suggest that domain-specific models can substantially outperform the proposed “universal” approach, directly challenging the paper’s central thesis.

3. The experimental results are muddled and fail to convincingly validate the proposed contributions. In the ablation studies, removing any one of the three core components—MA-CLIP, modality-aligned prompts, or DSRM—yields only marginal performance drops. This suggests that the modules may not be synergistically integrated, and their individual necessity is unclear. It raises the concern that the authors may have simply stacked components without identifying the essential mechanism driving performance gains.

**Questions:**

1. Why would modality-specific designs inherently be redundant? What fundamental commonalities exist across such diverse modalities (e.g., sparse events vs. dense depth vs. thermal signatures)? What justifies the existence of a single unified framework capable of handling all of them effectively? Given their vastly different data characteristics, what shared structure does your method exploit—and can you provide experimental evidence for this shared representation?

2. How does SegRGB-X compare against the actual SOTA methods in each domain? As noted, DFormer-v2 achieves 58.4 mIoU on NYUv2 and 67.1 mIoU on DELIVER—far surpassing your reported numbers. Doesn’t this demonstrate that specialized models still dominate? If so, what concrete advantages does your “universal” model offer over these domain-optimized approaches?

3. Given that ablating any single component barely affects performance (yet the full model is claimed to achieve SOTA), what is the true essence of your contribution? What is the intrinsic relationship among the three proposed modules? Could some of them be redundant? Please clarify which component(s) are truly responsible for performance gains and why all three are necessary.

---

### Official Review · Reviewer_dHgw · 2025-10-24

**Soundness:** 2
**Presentation:** 3
**Contribution:** 2
**Rating:** 4
**Confidence:** 4

**Summary:**

This paper introduces SegRGB-X, a generalist framework for semantic segmentation that aims to handle arbitrary combinations of RGB and an auxiliary modality. The proposed method features three key technical contributions: Modality-aware CLIP (MA-CLIP), Modality-aligned Embedding, and Domain-Specific Refinement Module (DSRM). The model is trained jointly on five diverse datasets covering event, thermal, depth, polarization, and light field modalities. The results show that SegRGB-X achieves a new state-of-the-art average mIoU of 65.03%, outperforming previous specialist models.

**Strengths:**

1. The integration of a pre-trained CLIP is interesting.
2. The authors conduct extensive experiments for validation.

**Weaknesses:**

1. Some components is not clearly introduced, such as how the modality-aligned embedding is designed. Another example is that hot input embeddings, control prompts from MA-CLIP, and modality-aligned prompts are combined is not introduced. Moreover, The text explicitly states there are "a total of four DSRM modules integrated throughout the network", yet the main architectural diagram (Figure 2) only shows a single DSRM in the final stage.
2. The model's performance on the NYUDepthV2 dataset is a significant concern. The authors attribute this to a domain gap, but this explanation undermines the central claim of building a "generalist" model.
3. Comparing a joint trained model with models trained on single dataset is unfair. The authors should compare under the same setting.
4. The proposed method leads to dramatic increase in computational cost with a relatively marginal improvement in mean mIoU. For many real-world applications, a significant slower model with slight higher performance is not a practical choice.

**Questions:**

see weakness

---

### Official Review · Reviewer_M2cX · 2025-11-01

**Soundness:** 2
**Presentation:** 2
**Contribution:** 2
**Rating:** 4
**Confidence:** 5

**Summary:**

This paper presents SegRGB-X, a versatile semantic segmentation model designed to work with any combination of RGB and an additional modality (RGB-X) within a single architecture. The main technical contributions are threefold. First, the authors introduce a Modality-Aware CLIP (MA-CLIP), which adapts CLIP to multiple modalities efficiently using LoRA. Second, they propose a modality-aligned embedding mechanism that employs learnable prompts to reduce the feature gap between CLIP and the backbone network. Third, a Domain-Specific Refinement Module (DSRM) is used to refine features in an adaptive way. SegRGB-X is trained and evaluated across five diverse datasets covering event, thermal, depth, polarization, and light field modalities. The results show that SegRGB-X consistently outperforms both specialist and generalist models in terms of mean IoU and maintains strong robustness under challenging conditions.

**Strengths:**

1. The framework is clearly illustrated in Figure 2, making the multi-component design (MA-CLIP, prompts, DSRM, etc.) easy to follow.
2. The feature map visualizations (Figure 8, Figure 10) across backbone stages provide convincing support for claims of modality-specific and hierarchical representation learning.

**Weaknesses:**

1. The model shows a significant drop in performance on NYUDepthV2 (Table 1: 47.77% vs. 56.93% or higher on other datasets). This appears to be more than a practical issue—it reflects a fundamental data imbalance, as most of the training data is outdoor while NYUDepthV2 is indoor (see Appendix A.5 on Generalization Error Analysis). Although the authors acknowledge this, the discussion could be strengthened by suggesting potential mitigation strategies, such as domain adaptation or balanced sampling techniques.
2. Overreliance on Specific Visual-Language Models: The approach relies solely on CLIP for vision-language alignment. By not evaluating alternative VLMs or more general unsupervised/self-supervised backbones (e.g., BLIP, ALIGN), the work may be limited in generality. While some justification is provided, direct experimental comparisons are lacking.
3. The label-space unification and remapping function described in Section 3.5 is an important component, particularly since datasets often differ semantically. It would be helpful if the authors clarified how overlapping or ambiguous labels are handled, and whether mismatches in class distribution affect either learning or evaluation.
4. The paper could clarify whether the joint model is trained using individual paired modalities or batch-wise paired modalities. It is also unclear whether all modalities are required simultaneously during inference, or if the model supports modality dropout. While some implementation details are provided in Section 4.1, the logic behind modality pairing and batch composition remains somewhat ambiguous.

**Questions:**

Please refer to the Weaknesses section.

---

### Note · Authors · 2025-11-14

I have read and agree with the venue's withdrawal policy on behalf of myself and my co-authors.